# Dietary Emulsifiers Exacerbate Food Allergy and Colonic Type 2 Immune Response through Microbiota Modulation

**DOI:** 10.3390/nu14234983

**Published:** 2022-11-23

**Authors:** Akihito Harusato, Benoit Chassaing, Charlène J. G. Dauriat, Chihiro Ushiroda, Wooseok Seo, Yoshito Itoh

**Affiliations:** 1Department of Gastroenterology, Kyoto Prefectural University of Medicine, Kajii-cho 465, Kamigyo-ku, Kyoto 602-8566, Japan; 2INSERM U1016, CNRS UMR8104, Université Paris Cité, Team «Mucosal Microbiota in Chronic Inflammatory Diseases», 24 rue du Faubourg Saint-Jacques Paris, 75014 Paris, France; 3Department of Clinical Nutrition, School of Medicine, Fujita Health University, Toyoake 470-1192, Japan; 4Department of Immunology, Nagoya University Graduate School of Medicine, Nagoya 464-0813, Japan

**Keywords:** food allergy, dietary emulsifiers, microbiota

## Abstract

The significant increase in food allergy incidence is correlated with dietary changes in modernized countries. Here, we investigated the impact of dietary emulsifiers on food allergy by employing an experimental murine model. Mice were exposed to drinking water containing 1.0% carboxymethylcellulose (CMC) or Polysorbate-80 (P80) for 12 weeks, a treatment that was previously demonstrated to induce significant alterations in microbiota composition and function leading to chronic intestinal inflammation and metabolic abnormalities. Subsequently, the ovalbumin food allergy model was applied and characterized. As a result, we observed that dietary emulsifiers, especially P80, significantly exacerbated food allergy symptoms, with increased OVA-specific IgE induction and accelerated type 2 cytokine expressions, such as IL-4, IL-5, and IL-13, in the colon. Administration of an antibiotic regimen completely reversed the emulsifier-induced exacerbated susceptibility to food allergy, suggesting a critical role played by the intestinal microbiota in food allergy and type 2 immune responses.

## 1. Introduction 

Food allergy triggers acute, potentially life-threatening hypersensitivity reactions, as well as chronic manifestations that mainly affect the gastrointestinal tract [1]. Currently, food allergy is recognized as a major health issue, since epidemiological studies revealed that the prevalence of food allergy is drastically increasing worldwide, particularly in modernized countries [2]. Although it is assumed genetic and environmental factors might play a key role in disrupting oral tolerance [3], to date, the etiology of food allergy remains to be elucidated, and once food allergy develops, no effective strategy to cure food allergy is established [4]. Therefore, secondary prevention to avoid allergens using elimination diets is currently the most efficient way to reduce the risk of food allergy-induced anaphylaxis [5].

On the other hand, dietary lifestyle has globally changed in the late 20th century, which is correlated with increased incidence of various chronic inflammatory diseases, metabolic syndrome, and food allergies [6,7]. This dietary change is characterized by a decreased consumption of unprocessed food consisting of natural components such as vegetables, fruits, and dietary fibers, and instead, by an increased intake of highly processed diets, such as cake mix, chocolate, and ice creams [8,9]. Among these highly processed diets, dietary emulsifiers are commonly used to stabilize texture, tastiness, and colors [10]. Acting as surfactants to avoid the separation of hydrophobic and hydrophilic components of processed foods, their addition also extends the product’s shelf life [9]. 

We and others formerly hypothesized that dietary emulsifiers might be correlated with the rapid increase in the incidence of chronic inflammatory diseases [8,11,12,13]. By administering universally used emulsifiers, polysorbate 80 (P80) and carboxymethylcellulose (CMC), at doses to model its general consumption, we observed changes in colonic inner mucus layer by promoting microbiota encroachment that induces chronic colitis in mice [8,14,15]. However, it is completely unknown whether these emulsifiers may affect the pathogenesis of food allergy-induced hypersensitivity reactions and type 2 immune responses. Importantly, recent studies have proposed that the intestinal microbiota is substantially involved in the etiology of food allergy, particularly in early life [16,17]. Therefore, in the present study, we aimed to elucidate whether exposure to dietary emulsifiers after neonatal periods significantly affects the pathogenesis of food allergy as well as the composition of the microbiota. To clarify the involvement of the microbiota, we employed 16 s rRNA sequencing analysis as well as the use of an antibiotic regimen in a murine model of OVA food allergy.

## 2. Materials and Methods

### 2.1. Mice

Three-week-old BALB/c mice were obtained from the Shimizu Laboratory Supplies Co., Ltd., Kyoto, Japan. Experiments were carried out using age and gender-matched groups. All animal experiments were implemented under the guidelines for the care and use of live animals of the Kyoto Prefectural University of Medicine (KPUM), and animal protocols were approved by the Institutional Animal Care and Use Committee of KPUM (Approval Code: M2022-514). All mice were kept at 18–24 °C with 40–70% humidity and maintained under a 12 h/12 h dark/light cycle. 

### 2.2. Emulsifier Agent Treatment

Sodium carboxymethylcellulose (CMC, average Mw ~250,000) and polysorbate-80 (P80) were purchased from Sigma (Sigma, St. Louis, MO, USA). Mice were exposed to CMC and P80 diluted in drinking water (1.0%). The equivalent water (Kyoto city water) was used for the water (control) group. Fresh feces were collected at the beginning and the end of experiments for next-generation sequencing analysis.

### 2.3. Food Allergy Induction

After six weeks of exposure to CMC and P80, mice were sensitized with intraperitoneal injection with 20 mg of ovalbumin followed by another sensitization in two weeks. After another two weeks, the mice were orally administered 50 mg of ovalbumin a total of five times. Oral administration was performed with intragastric feeding needles (Cat No. 6202, Fuchigami, Kyoto, Japan). Treatment groups were defined as follows: Control (sham); Water (food allergy with water treatment); CMC (food allergy with CMC treatment); P80 (food allergy with P80 treatment). 

Following the final OVA administration, diarrhea scores were assessed by a severity score of the stool forms (0, solid-state; 1, funicular; and 2, slurry or watery). In addition, rectal temperature was measured by exploiting a weighing environment logger (AD-1687, A&D Co., Ltd., Tokyo, Japan). Mice were then euthanized and organs and blood were collected for downstream analysis. Blood plasma was gathered by centrifugation and stored at −80 °C for ELISA. In some experiments, mice were provided ampicillin (1 g/L), vancomycin (500 mg/L), neomycin sulfate (1 g/L), and metronidazole (1 g/L) in drinking water from five days prior to beginning the OVA challenge.

### 2.4. Quantitative Real-Time PCR

RNA was isolated using the acid-guanidinium-phenol-chloroform method from the samples preserved in TRIzol (Thermofisher, Waltham, MA, USA). Reverse transcription was implemented using a high-capacity cDNA Reverse Transcription Kit (Applied Biosystems, Waltham, MA, USA) and quantitative real-time PCR was performed using a 7300 Real-Time PCR system with Power SYBR Green PCR Master Mix (Applied Biosystems).

Primers used were: 

mActb (F, TATCCACCTTCCAGCAGATGT; R, AGCTCAGTAACAGTCCGCCTA)

mIl4 (F, CGCCATGCACGGAGATG; R, CGAGCTCACTCTCTGTGGTGTT)

mIl5 (F, TGACCGCCAAAAAGAGAAGTGT; R, ACTCTTGCAGGTAATCCAGGAACT)

mIl13 (F, CGCAAGGCCCCCACTA; R, AAAGTGGGCTACTTCGATTTTGG)

mIl1f9 (F, AGAGTAACCCCAGTCAGCGTG; R, AGGGTGGTGGTACAAATCCAA)

### 2.5. ELISA

OVA-specific IgE antibody levels were determined using ELISA kits (DS Pharma Biomedical Co., Ltd., Chuo-ku, Osaka, Japan; Chondrex, Inc., Woodinville, WA, USA) according to the manufacturer’s protocol. Blood plasma (diluted to 1:100) was incubated with OVA at 4 °C overnight. Then secondary antibody solution, streptavidin peroxidase solution, and TMB solution were serially added. OVA-specific IgE antibody levels were then quantified at 450 nm/630 nm.

### 2.6. 16S rRNA Sequencing 

As formerly described, 16S rRNA gene sequencing was implemented [8]. Briefly, quality filtering was performed using QIIME2, and sequences were aligned to operational taxonomic units (OTUs) using the UCLUST algorithm and classified taxonomically using the Greengenes reference database 13_8. Principal coordinates analysis (PCoA) plots were used to assess the variation between experimental groups (beta diversity). 

### 2.7. Isolation of Colonic Lamina Propria (LP) Cells

Isolation of colonic LP cells was performed as previously described [18]. Briefly, colonic tissue was cut into pieces 0.5 cm in length and placed in an orbital shaker for 20 min at 37 °C in HBSS with 10% FCS and 2 mM EDTA. This shaking step was repeated and then the remaining tissue was minced and placed in HBSS with 10% FCS, 1 mg/mL collagenase IV (Sigma), and 40 U/mL DNase (Roche) I and shaken for 20 min at 37 °C. The contents are then filtrated through a 70 μm cell strainer directly into a 50 mL conical tube. Each 50 mL conical tube is topped off with HBSS/FBS and centrifuged at 4 °C. After the repetition of this step, the supernatant is poured off and the cell pellet is resuspended in ice-cold HBSS/FBS. From these resulting LP cells, CD45+ immune cells were isolated by using anti-CD45 magnetic beads (Miltenyi) and subsequently used for downstream analysis.

### 2.8. RNA-seq

The RNA was isolated from CD45+ colonic LP cells by utilizing an RNA isolation kit (Qiagen, Hilden, Germany) at Nagoya University. The RNA-Seq experiments were performed by Macrogen. Library construction was performed using TruSeq stranded mRNA library preparation kit. Sequencing was implemented on the Illumina platform (NovaSeq6000). Volcano plots and heat maps were generated using R.

### 2.9. Histology

Intestinal tissues were fixed in 10% formalin. Paraffin embedding, sectioning, and Periodic Acid Schiff (PAS) staining were performed at Kyoto Micro Bio Laboratory. Two to four nanometer-tissue sections were stained with PAS for mucins in goblet cells. PAS-positive cells were quantified, as formerly reported [19].

### 2.10. Statistics

Statistical analyses were performed by using GraphPad Prism software, version 9.0 (Graphpad Software). Student’s *t*-test or One-way ANOVA and Tukey’s Multiple Comparison Test were used to determine significance. * *p* < 0.05, ** *p* < 0.01, *** *p* < 0.001, **** *p* < 0.0001; ns, not significant.

## 3. Results

### 3.1. Dietary Emulsifier-Treated Mice Showed Symptoms of Severe Food Allergy

With the goal to investigate the contribution of dietary emulsifier exposure in the development of food allergy, three-week-old BALB/c mice were exposed to drinking water containing 1.0% carboxymethylcellulose (CMC) or Polysorbate-80 (P80) for 12 weeks. Oral allergen-induced food allergy was provoked by intraperitoneal OVA administration with alum after six weeks of emulsifier exposure, followed by oral challenges with OVA, as described in Figure 1A. In this food allergy model, hypothermia, as a sign of anaphylaxis as well as watery diarrhea are observed in the mice treated with water, while the control mice (sham) kept their body temperature at a steady state, and defecation was normal (Figure 1B,C). Next, when compared with the mice treated with water or CMC, the mice treated with P80 showed more severe hypothermia (Figure 1B). Indeed, P80-treated mice developed significantly more severe diarrhea as compared to the mice treated with water or CMC (Figure 1C). Furthermore, the blood level of OVA-specific IgE was upregulated in P80-treated mice as compared to water or CMC-treated mice (Figure 1D). These results suggest that the treatment with P80 significantly exacerbates food allergen-induced allergy and anaphylaxis in the OVA-induced food allergy model.

### 3.2. Colonic Type 2 Immune Response Is Exacerbated by Dietary Emulsifier Consumption

Although food allergy is induced by orally exposed allergens, the section of the gastrointestinal tract contributing to the allergic response, especially in the setting exposed to dietary emulsifiers, remains unknown. To clarify this point, we examined gene expressions of type 2 cytokines in the duodenum, jejunum, ileum, and colon segments. We first examined the gene expressions of IL-4, IL-5, and IL-13 in water-treated mice with or without food allergy. Intriguingly, the expressions of these type 2 cytokines were relatively higher in colonic tissue as compared with tissues derived from other parts of the GI tract (Appendix A), indicating that type 2 immune response in the colon could play a substantial role in the development of food allergy and anaphylaxis. Hence, we next investigated the gene expressions of these cytokines in emulsifier-treated mice subjected to food allergy. Surprisingly, among the three groups studied (water, CMC, and P80), we did not detect significant differences in the expressions of these cytokines in the duodenum, jejunum, and ileum (Figure 2A–C). More importantly, we observed that IL-4 and IL-5 gene expressions were higher in the colon of P80-treated mice compared to water-treated animals (Figure 2A–C), suggesting that colonic immune response may contribute to the exacerbated allergic response in P80- treated mice. 

### 3.3. Periodic Acid-Schiff Staining of GI Tissue

We next investigated the levels of goblet cell (GC) hyperplasia by performing periodic acid-Schiff (PAS) staining of duodenal and colonic tissue. In water-treated mice with food allergy, the PAS staining revealed that GC hyperplasia was significantly accelerated in the colon as compared with that in the duodenum of the mice with food allergy (Appendix A). Moreover, severe GC hyperplasia was observed in the colon of dietary emulsifier-treated mice, both CMC and P80 groups, as presented in Figure 3A,B. These results further support that allergic response in the colonic segment of the gastrointestinal tract is playing a role in the emulsifier-induced exacerbation of food allergy and anaphylaxis. 

### 3.4. Dietary Emulsifiers Facilitate Mast Cell Activation and Type 2 Immune Responses in Colonic Immune Cells

Based on the data described above regarding colonic immune responses to food allergy and in the context of dietary emulsifier exposure, we next investigated to which extent dietary emulsifiers consumption affected the landscape of allergy-related genes, by performing RNA-seq analysis of isolated CD45+ immune cells in the colon (Figure 4A–C). As a result, dietary emulsifiers, and especially P80, significantly impacted the expression level of numerous genes in the CD45+ immune cell population, as observed using Volcano plot representation (Figure 4A,B). Dysregulations include upregulation in the expressions of genes involved in mast cell activation (Cpa3, Mcpt1, Mcpt2, Mcpt5, Tph1), histamine synthesis (Hdc), type 2 cytokines (IL-4, IL-5, IL-13, IL-33). Additionally, gene expressions related to neutrophil, monocyte, and eosinophil recruitment (Cxcl5, Ly6g, Ccl1, Ccl2, Ccl11, Ccl24) were also significantly increased in P80-treated mice compared to controls. Of note, we also found that IL-1f9 (IL-36γ), an IL-1 cytokine family member, was significantly upregulated in the samples derived from dietary emulsifier-treated mice compared to water-treated mice. Indeed, as compared with control mice (sham), qPCR analysis confirmed that IL-36γ was induced in food allergy (Appendix A). The induction of IL-36γ was especially higher in colonic tissue samples, and further enhanced by P80 treatment (Appendix A), hence consistent with the RNA-seq analysis. Altogether, these findings further validate dietary emulsifier-mediated exacerbation of allergic and type 2 immune responses in the colon.

### 3.5. Alteration of the Microbiota in Dietary Emulsifiers Treated Mice with Food Allergy

Since several reports using C57BL/6 mice and humans suggest that dietary emulsifier treatment significantly affects microbiome composition [8], we examined microbiota composition in our food allergy model through 16S rRNA gene sequencing. While we did not find any differences in microbiota composition between the three groups before the treatment, we observed stark alterations in microbiota composition following P80 consumption (week 12, Figure 5A,B). We also investigated microbiota composition in the mesenteric lymph nodes using 16S rRNA seq analysis, an approach that revealed no significant differences between water-treated and emulsifier-treated mice (unpublished observation). Altogether, these results indicate an altered microbiota composition induced by P80 consumption during food allergy development, which may be involved in exacerbating food allergy and colonic type 2 immune responses. 

### 3.6. Elimination of Microbiota Reversed Dietary Emulsifier-Mediated Exacerbated Susceptibility to Food Allergy

In order to clarify the role played by the intestinal microbiota in exacerbating susceptibility to food allergy in emulsifier-treated mice, we next implemented experiments aiming to deplete microbiota, through the administration of an antibiotic regimen for five days prior to beginning the OVA challenge (Figure 6A). We importantly observed that administration of such an antibiotic regimen was sufficient to completely reverse parameters of allergy-induced anaphylaxis in emulsifier-treated mice (Figure 6B,C). Moreover, when colonic goblet cells were analyzed, we did not observe any differences between water-treated mice and dietary emulsifier-treated mice after the administration of the antibiotic cocktail (Figure 6D,E). These findings importantly revealed a critical role played by the intestinal microbiota in aggravating food allergy and type 2 immune responses in dietary emulsifier-treated mice.

## 4. Discussion

Among food additives, emerging evidence from mice and human studies indicates that dietary emulsifiers induce microbiota dysbiosis by promoting disease development including chronic inflammatory disorders, metabolic syndrome, and cancer [8,14,20]. These findings highlight the crucial role of dietary constituent-mediated modulation of the intestinal environment in health and disease [21]. However, it has not been studied that such dietary emulsifiers could affect the development of allergic reactions and type 2 immunity. Therefore, we explored whether dietary emulsifiers are contributing to the pathogenesis of food allergy by using an established model of OVA-induced food allergy. Our findings demonstrated that the dietary emulsifiers, especially P80, induced severe food allergy, with enhanced parameters of anaphylaxis including diarrhea and hypothermia. Since it has not been clearly defined which segment of the intestine is involved in food allergy-induced type 2 immune responses, we investigated the gene expressions of IL-4, IL-5, and IL-13 in the duodenum, jejunum, ileum, and colon by using quantitative PCR. As a result, the gene expressions of these cytokines were relatively higher in the colon, as compared with other parts of the intestine. Further, we confirmed that GC hyperplasia, which is an important feature of allergy-induced mucus overproduction, was even aggravated in the colon as compared with that in the duodenum by PAS staining. Although a previous report has shown that enhanced allergic responses were observed in the colon as well as in the duodenum [22], these results suggest that allergic responses in the colon are substantially contributing to developing the symptom of anaphylaxis in our setting.

To further validate if dietary emulsifiers modulate food allergy-induced type 2 immune responses, we implemented the RNA-seq analysis for isolated colonic CD45+ immune cells from the mice-induced food allergy. Consequently, P80 significantly enhanced type 2 inflammatory pathways followed by the activation of mast cells related genes, although CMC also enhanced those pathways the extent was relatively mild. Interestingly, we also found that the expression of a novel IL-1 family member, IL-36γ was significantly upregulated in P80 treated group. Although the role of IL-36 cytokines in mucosal immunity has been reported so far [23,24], to our knowledge, this is the first study showing the involvement of IL-36 in food allergy. 

We have so far reported that dietary emulsifiers modulate microbiota composition in C57BL/6 mice and human studies [8,25]. Consistent with these findings, 16S rRNA analysis of fecal microbiota showed that a dietary emulsifier P80 significantly altered microbiota, while CMC did not affect microbiota in a food allergy model, suggesting the specific correlation between P80-mediated modulation of the microbiota and allergic immune response. In addition, the administration of antibiotics to eliminate microbiota, totally suppressed the aggravated allergic reaction in emulsifier-treated mice. These data suggest that communications between colonic immune cells and microbiota could play a key role in the pathogenesis of food allergy. Since the onset of food allergy usually occurs in the early stage of life [2], it is quite possible that environmental elements are involved in the mechanism of food allergy development [6]. Indeed, epidemiologic studies have shown maternal dietary habits, sunlight, microbiome, and environmental allergens are involved in the development of food allergy [26,27,28,29]. Therefore, it is possible that exposure to dietary components, such as emulsifiers in early life, or even during maternal pregnancy could affect the composition of the microbiota and the development of food allergy. Importantly, microbial-derived metabolites such as G protein-coupled receptor ligands, aryl hydrocarbon receptor ligands, and tryptophan metabolites may regulate immune responses [30,31,32]. Accordingly, we previously reported that dietary emulsifiers altered fecal levels of short-chain fatty acids and bile acids [8]. Thus, given the profound influence of dietary emulsifiers in altering microbiota and its metabolites, and accelerating food allergy, targeting the interactions played by colonic immune cells and microbiota may hold great promise to control dysregulated type 2 immune responses. Additionally, our data from this study propose the significance of colonic type 2 immunity in the pathogenesis of food allergy. Further investigations are warranted to elucidate the importance of the intestinal microbiota and type 2 immunity in the gut to the development of food allergy.

## Figures and Tables

**Figure 1 nutrients-14-04983-f001:**
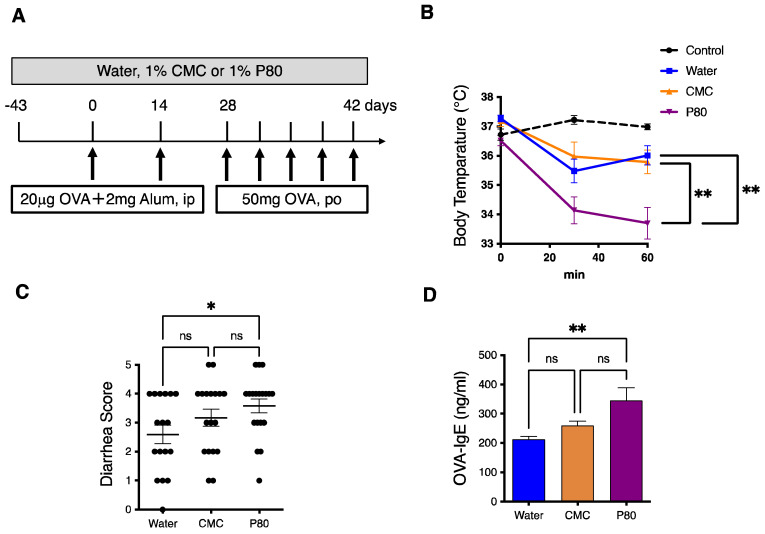
Dietary emulsifiers exacerbated food allergy-induced parameters of anaphylaxis. (**A**) Experimental flow chart: three-week-old BALB/c mice were treated with Water, CMC, or P80 for three months. After six weeks of exposure to Water, CMC, or P80, mice were sensitized with intraperitoneal injection with 20 mg of OVA and Alum, followed by another sensitization in two weeks. After another two weeks, the mice were orally administered 50 mg of ovalbumin five times. Mice were euthanized at the end of experiments and used for subsequent analysis. (**B**) Time course of hypothermia. Rectal temperature was evaluated just after the final oral challenge with OVA (*n* = 5, Control; *n* = 10, Water; *n* = 9, CMC; *n* = 10, P80). (**C**) Diarrhea scores. The extent of diarrhea was examined just after the final oral challenge with OVA (*n* = 17, Water; *n* = 18, CMC; *n* = 19, P80). (**D**) OVA-specific IgE in plasma collected from food allergy-induced mice was measured by ELISA (*n* = 10 per group). One-way ANOVA and Tukey’s Multiple Comparison Test were used to determine significance. Data are the means ± SEM. * *p* < 0.05, ** *p* < 0.01; ns, not significant.

**Figure 2 nutrients-14-04983-f002:**
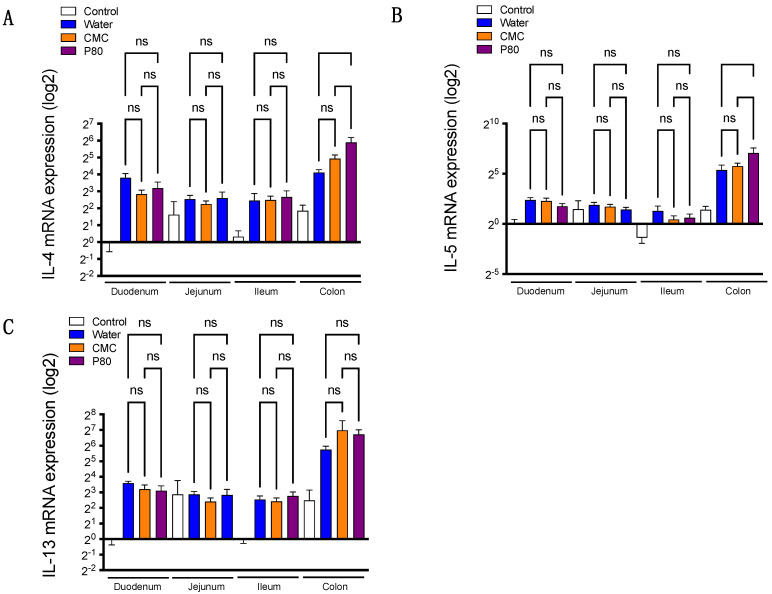
(**A**–**C**) The mRNA expression for IL-4, IL-5, and IL-13 in duodenum, jejunum, ileum, and colon by using quantitative PCR. The white bar represents the control group without food allergy induction; the blue bar represents the food allergy-induced group treated with water (no dietary emulsifier treatment); orange bar represents CMC treated group; the purple bar represents P80 treated group. (*n* = 5, Control; *n* = 10, Water; *n* = 9, CMC; *n* = 9, P80). One-way ANOVA and Tukey’s Multiple Comparison Test were used to determine significance. Data are the means ± SEM. ns, not significant.

**Figure 3 nutrients-14-04983-f003:**
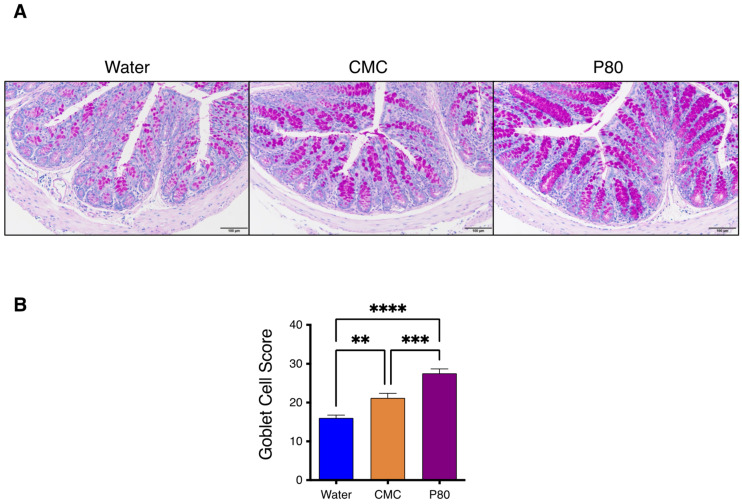
(**A**) Periodic acid-Schiff (PAS) staining for colonic tissue in food allergy-induced mice (Water, CMC, P80). (**B**) Goblet Cell Score of PAS-stained colonic tissue as of A. (*n* = 18 per group) One-way ANOVA and Tukey’s Multiple Comparison Test were used to determine significance. Data are the means ± SEM. ** *p* < 0.01, *** *p* < 0.001, **** *p* < 0.0001.

**Figure 4 nutrients-14-04983-f004:**
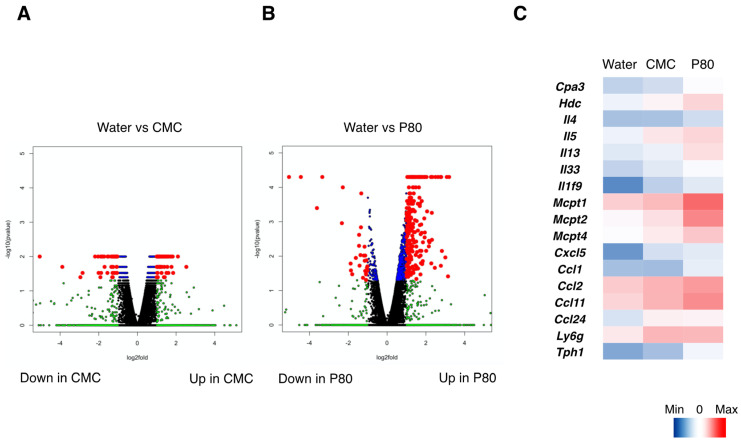
RNA-seq analysis of colonic CD45+ immune cells derived from the mice with food allergy. (**A**) Volcano plots presenting log2 fold-change of gene expression (X axis), and log10 fold-change of gene expression (Y axis) in CMC-treated mice as compared with water-treated mice. Blue dots correspond to the genes with a *p* value < 0.05 between CMC-treated and water-treated mice. Green dots correspond to the genes with at least a 2-fold-decreased or 2-fold-increased abundance in CMC-treated mice compared to water-treated mice. Red dots correspond to the genes with at least a 2-fold-decreased or 2-fold-increased abundance in CMC-treated mice compared to water-treated mice and a *p* value < 0.05. (**B**) Volcano plots presenting log2 fold-change of gene expression (X axis), and log10 fold-change of gene expression (Y axis) in P80-treated mice as compared with water-treated mice. Blue dots correspond to the genes with a *p* value < 0.05 between P80-treated and water-treated mice. Green dots correspond to the genes with at least a 2-fold-decreased or 2-fold-increased abundance in P80-treated mice compared to water-treated mice. Red dots correspond to the genes with at least a 2-fold-decreased or 2-fold-increased abundance in P80-treated mice compared to water-treated mice and a *p* value < 0.05. (**C**) Heatmaps showing the gene expressions related to mast cell activation and type 2 immune responses.

**Figure 5 nutrients-14-04983-f005:**
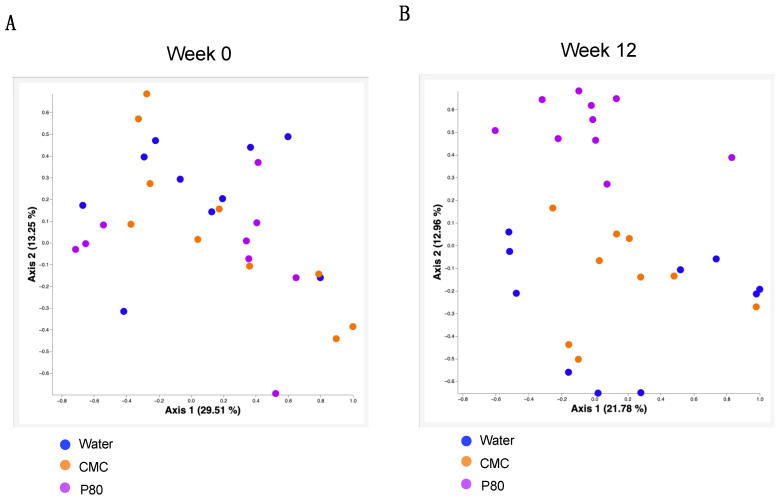
Alteration of microbiota in dietary emulsifiers treated mice with food allergy are expressed by Principal coordinates analysis (PCoA) plots to assess the variation among water, CMC, and P80 treated group at week 0 (**A**) and week 12 (**B**).

**Figure 6 nutrients-14-04983-f006:**
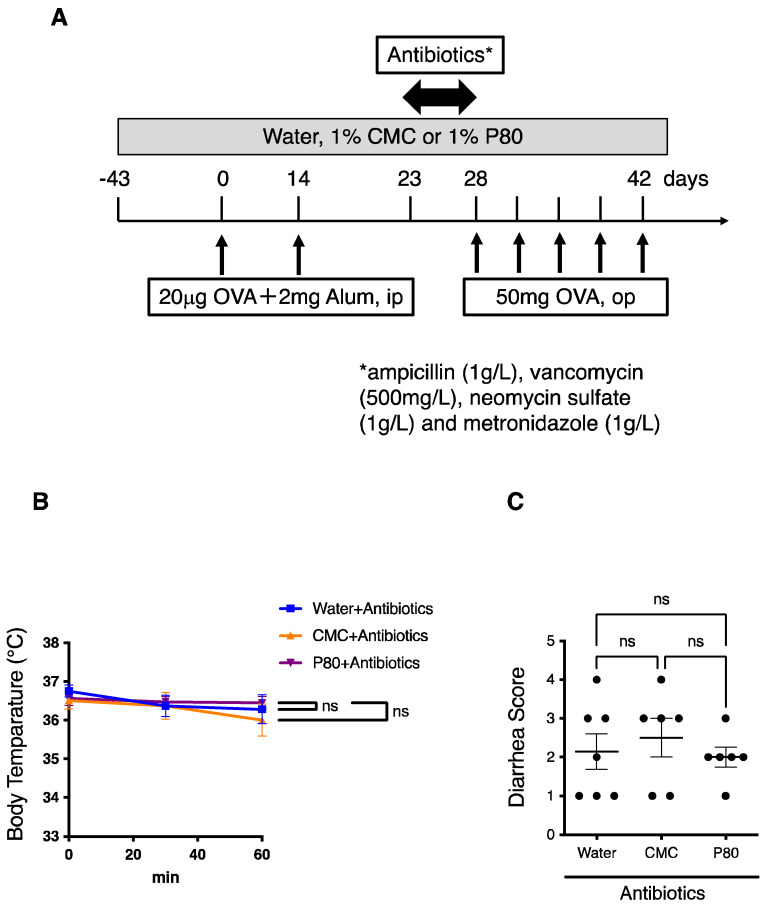
Treatment with an antibiotics regimen completely reversed aggravated parameters of food allergy. (**A**) Experimental flow chart. (**B**) Time course of hypothermia. Rectal temperature was evaluated just after the final oral challenge with OVA (*n* = 7, Water; *n* = 6, CMC; *n* = 6, P80). (**C**) Diarrhea scores. The extent of diarrhea was examined just after the final oral challenge with OVA (*n* = 7, Water; *n* = 6, CMC; *n* = 6, P80). (**D**) PAS staining for colonic tissue in food allergy-induced mice treated with an antibiotic regimen plus, water, CMC, or P80. (**E**) Goblet Cell Score of PAS-stained colonic tissue as of D. (*n* = 12 per group). One-way ANOVA and Tukey’s Multiple Comparison Test were used to determine significance. Data are the means ± SEM. * *p* < 0.05; ns, not significant.

## Data Availability

16S rRNA sequencing and RNA-seq data are available upon request.

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
