# Peer review of "Dietary Emulsifiers Exacerbate Food Allergy and Colonic Type 2 Immune Response through Microbiota Modulation"

_nutrients, 2022, doi:10.3390/nu14234983_

Round 1

Reviewer 1 Report

Reviewer comments:

The manuscript entitled “Dietary emulsifiers exacerbate food allergy and colonic type immune response through microbiota modulation” investigated the impact of dietary emulsifiers on food allergy by using mice model. The authors suggested that the dietary emulsifiers intake significantly exacerbated food allergy, with increased severity of diarrhea, hypothermia and accelerated type 2 immune responses in colon. While, Administration of an antibiotic regimen completely reversed the emulsifier-induced exacerbated susceptibility to food allergy. The present study confirmed the significant role of microbiota in food allergy and type 2 immune responses in mice exposed to dietary emulsifiers.

Generally, the present work is of good quality, and have provided some interesting results that might be helpful for uncovering the role of gut microbiota in food allergy response. However, there are also some problem need to be addressed before being further processed.

1. The abstract part is suggested to be improved. The present version just descriptive introduced the work carried out in the present manuscript, while there was no data for supporting these results. More quantitative data should be added.

2. In the introduction part, the correlations between the gut microbiota and food allergy response should be mentioned, and more related references can be cited here.

3. The last paragraph (Line 54-65) of the introduction part is suggested to remove, for that is the results of the present work, rather than a part of ‘introduction’.

4. The aims, backgrounds, and the main work of the present study should be introduced in the end of the introduction part. The present introduction is not a suitable one and should be improved.

5. Line 69-70. The authors used the 3 weeks-old BALB/c mice for experiment, that is not in conformity with the ethics of experimental animals. For the mice reach in adulthood in 6-8 week, the authors should provide explanations for using the baby mice for experiments. The authors are also should provide ethic statement in the materials and methods part in line 251-257, and the necessary documents for animal experiment also should be provided.

6. The “Figure” used in the Figure captions is not in accordance with that referred in the main context of the manuscript (e.g. Fig. 1, Fig. 2 etc.). Please make revisions.

7. The template used for the present manuscript seemed not the original one (including the size of figures, the fonts of the figure captions), please check. When using the template, please do not change the format.

8. The results part is good, but the discussion part should be strengthened. More references should be cited. The authors are suggested to focused on discussion about the topic related with the content of the current work, and provide in depth discussion.

9. Line 294-298. The format should be improved.

10. Line 290-293, Line 311-317. The briefly description of the operation procedures of the Elisa, cell RNA-seq and tissue histology experiments are suggested to be provided.

11. The overall language should be improved. some grammar mistakes and language errors should be revised. please have double check. 

Author Response

Reviewer 1:

The manuscript entitled “Dietary emulsifiers exacerbate food allergy and colonic type immune response through microbiota modulation” investigated the impact of dietary emulsifiers on food allergy by using mice model. The authors suggested that the dietary emulsifiers intake significantly exacerbated food allergy, with increased severity of diarrhea, hypothermia and accelerated type 2 immune responses in colon. While, Administration of an antibiotic regimen completely reversed the emulsifier-induced exacerbated susceptibility to food allergy. The present study confirmed the significant role of microbiota in food allergy and type 2 immune responses in mice exposed to dietary emulsifiers.

Generally, the present work is of good quality, and have provided some interesting results that might be helpful for uncovering the role of gut microbiota in food allergy response. However, there are also some problem need to be addressed before being further processed.

Specific comments:

  1. The abstract part is suggested to be improved. The present version just descriptive introduced the work carried out in the present manuscript, while there was no data for supporting these results. More quantitative data should be added.

Thank you so much for the comments. According to the suggestion from the reviewer, we have added more quantitative data, such as OVA-specific IgE, IL-4, IL-5 and IL-13 into the abstract.

  1. In the introduction part, the correlations between the gut microbiota and food allergy response should be mentioned, and more related references can be cited here.

Thank you very much for pointing this out. We have added the sentence regarding the correlations between the gut microbiota and food allergy response, as well as related references, into introduction.

  1. The last paragraph (Line 54-65) of the introduction part is suggested to remove, for that is the results of the present work, rather than a part of ‘introduction’.

We truly appreciate the suggestion, and we have removed this part accordingly.

  1. The aims, backgrounds, and the main work of the present study should be introduced in the end of the introduction part. The present introduction is not a suitable one and should be improved.

Thank you for the important suggestion. We have added the sentences at the end of introduction, accordingly.

  1. Line 69-70. The authors used the 3 weeks-old BALB/c mice for experiment, that is not in conformity with the ethics of experimental animals. For the mice reach in adulthood in 6-8 week, the authors should provide explanations for using the baby mice for experiments. The authors are also should provide ethic statement in the materials and methods part in line 251-257, and the necessary documents for animal experiment also should be provided.

We thank the reviewer for pointing out this important point. As a consequence, we have added details at the end of the revised introduction section in order to explain that early-life is playing a critical role in the etiology of food allergy as well as microbiota establishment. It is indeed well known that the prevalence of food allergy is particularly higher in early childhood. Moreover, early life environment may affect the condition of individuals later in adulthood. Therefore, many studies used those animal models. (Cf. Anna MR Hayes et al. Early Life Low-Calorie Sweetener Consumption Impacts Energy Balance during Adulthood, Nutrients 2022) This is the scientific reason we employed the mice at younger ages. Regarding the ethics for animal experiments, we have actually provided the information for the editorial office, which is shown as below. We included the approval code into method part.

Our study was approved by the KPUM Institutional Animal Care and Use Committee.

This committee actually deals with ethics in animal experiments.

Ethic Committee Name:  the KPUM Institutional Animal Care and Use Committee of KPUM

Approval Code: M2022-514

Approval Date: 2022 May 1st

Furthermore, we would like to provide the contents of the document as below, which were translated from Japanese into English.

Using BALB/c mice, a food allergy disease model is created by administering OVA albumin (intraperitoneal and gavage administration) to clarify the onset mechanism of food allergy. The breeding period is 6 to 18 weeks, the experiment is conducted with 8 animals per group, and both food and water are given ad libitum during the breeding period. During the breeding period, fecal collection and a small amount of blood collection from the tail vein are performed. In all procedures, isoflurane anesthesia was used to reduce pain, and after the end of the test period, the animals were euthanized without pain by overdosing barbiturate anesthetics. Histological, molecular biological and immunological examinations are performed. In the case of severe allergic symptoms such as shock are observed, or if abnormalities such as rapid weight loss of 20% or more are observed, euthanasia by overdose of barbiturate anesthetic are implemented as a humanitarian endpoint. All carcasses after the end of the experiment will be incinerated at the Experimental Animal Center.

  1. The “Figure” used in the Figure captions is not in accordance with that referred in the main context of the manuscript (e.g. Fig. 1, Fig. 2 etc.). Please make revisions.

We appreciate the important comment. We have made revisions accordingly.

  1. The template used for the present manuscript seemed not the original one (including the size of figures, the fonts of the figure captions), please check. When using the template, please do not change the format.

Thank you for pointing this out. We would like to upload the original figures as well.

  1. The results part is good, but the discussion part should be strengthened. More references should be cited. The authors are suggested to focused on discussion about the topic related with the content of the current work, and provide in depth discussion.

Thank you so much for the comments. We have added the sentences regarding the related topic, into discussion part.

  1. Line 294-298. The format should be improved.

Thank you for finding this out. The current format is produced by the editorial office. We would like to request the editorial office to be fixed.

  1. Line 290-293, Line 311-317. The briefly description of the operation procedures of the Elisa, cell RNA-seq and tissue histology experiments are suggested to be provided.

We appreciate the comments. We have added brief description for the Elisa, cell RNA-seq and tissue histology experiments, respectively.

  1. The overall language should be improved. some grammar mistakes and language errors should be revised. please have double check.

Thank you for the comments. We have double checked grammar and language.

Reviewer 2 Report

Manuscript ID: nutrients-2031892

Title: Dietary emulsifiers exacerbate food allergy and colonic type 2 immune response through microbiota modulation

Authors: Akihito Harusato * , Benoit Chassaing * , Charlène Dauriat , Chihiro Ushiroda , Wooseok Seo , Yoshito Itoh

Overview and general recommendation:

In the manuscript authors were investigating the impact of dietary emulsifiers on food allergy by employing an experimental murine model. Although the subject of the manuscript deals with the very important issue of food allergies caused by additives used in food production, several issues raised in the manuscript require improvement

Below I give some comments:

Major comments:

1.     Maybe it is worth adding information to the Introduction on the role of the microbe in the occurrence of allergies? The authors mention the importance of this factor in Results, but the Introduction has no theoretical basis.

2.     There is no clearly defined research aim, formulated e.g. at the end of the Introduction, please formulate it.

3.     Please, preferably in the methodology, define what is the control group and which group is Water? Some analysis (and methodology) have Water as a control group in others Control is a separate group from Water? - requires an clarification, especially since differences are indicated between the control group and the water.

Minor comments:

1.     Line 24: “Food Allergy,; Dietary Emulsifiers; MIcrobiota”- typing errors

2.     Introduction -lines 49-50 - Can the mechanism by which these two different compounds (P80, CMC) influence the induction of food allergies be determined / described?

3.     Lines 53-54: “Here, we demonstrated that mice treated with dietary emulsifiers are more susceptible to food allergy when compared to the control (water treated mice).” - What does "here" refer to or the research in this manuscript? or to the previously cited studies?

4.     Lines 242-248 - Can such strong conclusions be drawn from just one experiment? Shouldn't it be added that the research shows a promising trend, but that more research should be done in this direction?

5.     The text got unformatted in some places (eg. methodology)

6.     Line 318 – Statistic - many tests were proposed, the results of which were statistically compiled, but it is not known which method was used for which analysis?

7.     In the text, reference numbers should be placed in square brackets [ ]

Author Response

Reviewer 2:

Overview and general recommendation:

In the manuscript authors were investigating the impact of dietary emulsifiers on food allergy by employing an experimental murine model. Although the subject of the manuscript deals with the very important issue of food allergies caused by additives used in food production, several issues raised in the manuscript require improvement

Below I give some comments:

Major comments:

  1. Maybe it is worth adding information to the Introduction on the role of the microbe in the occurrence of allergies? The authors mention the importance of this factor in Results, but the Introduction has no theoretical basis.

We truly appreciate the comments. We have added the sentence and the related references regarding the role of the microbe in the occurrence of allergies. We have also amended the introduction part accordingly.

  1. There is no clearly defined research aim, formulated e.g. at the end of the Introduction, please formulate it.

Thank you for pointing this out. We have added those sentences at the end of introduction part.

  1. Please, preferably in the methodology, define what is the control group and which group is Water? Some analysis (and methodology) have Water as a control group in others Control is a separate group from Water? - requires an clarification, especially since differences are indicated between the control group and the water.

We are grateful to the reviewer for this important comment. We defined ‘Control’ as no food allergy induction (sham). We defined ‘Water’ as the food allergy induced mice treated with water. This point was clarified in our revised manuscript by defining treatment groups in the method section.

Minor comments:

  1. Line 24: “Food Allergy,; Dietary Emulsifiers; MIcrobiota”- typing errors

We have amended this part. This could be happened during the submission process.

  1. Introduction -lines 49-50 - Can the mechanism by which these two different compounds (P80, CMC) influence the induction of food allergies be determined / described?

Thank you for pointing this out. We have added the sentence into discussion section, to describe the possible mechanism, how the two different compounds (P80, CMC) influenced the induction of food allergies and microbiota.

  1. Lines 53-54: “Here, we demonstrated that mice treated with dietary emulsifiers are more susceptible to food allergy when compared to the control (water treated mice).” - What does "here" refer to or the research in this manuscript? or to the previously cited studies?

Thank you for the comments. As the reviewer1 also pointed out, we have removed this part from the manuscript.

  1. Lines 242-248 - Can such strong conclusions be drawn from just one experiment? Shouldn't it be added that the research shows a promising trend, but that more research should be done in this direction?

Thank you so much for the comments. We have modified the conclusion, which is not as strong as the previous one. We fully agree that the conclusion was too strong, and more detailed studies are necessary.

  1. The text got unformatted in some places (eg. methodology)

Thank you for finding this out. The current format is produced by the editorial office. We would like to request the editorial office to be fixed.

  1. Line 318 – Statistic - many tests were proposed, the results of which were statistically compiled, but it is not known which method was used for which analysis?

We appreciate the comments. Accordingly, we have added the use of statistical analysis into each figure legend.

  1. In the text, reference numbers should be placed in square brackets [ ]

Thank you for pointing this out. We have modulated the format for brackets, consistently.
